# AN ENSEMBLE LEARNING FRAMEWORK FOR VISIBILITY PREDICTION IN INDO-GANGETIC REGION

**Arkapal Panda**
Computer Vision and Pattern Recognition Unit
Indian Statistical Institute Kolkata, India
arkapalpanda88in@gmail.com

**Vaibhav Kumar** *& **Tanmay Basu**
Department of Data Science and Engineering,
Indian Institute of Science Education & Research
Bhopal, India
{tanmay,vaibhav}@iiserb.ac.in

## ABSTRACT

Visibility of an area can affect all forms of transportation and hence it is important to accurately estimate the visibility of an area for the upcoming days based on different parameters of the meteorological data to take precautions. Several machine learning techniques have been already applied on different kinds of data sets to estimate the visibility, however, none of them were explored on the Indo-Gangetic plane, which witnesses widespread fog primarily during winter. In this spirit, a regression framework is developed to estimate the visibility of the Indo-Gangetic region using the meteorological data, which outperforms the state of the arts.

## 1 INTRODUCTION

Visibility conditions depend mainly on different phenomena such as changes in air-mass, winds, temperature profiles etc. These phenomena in turn depend on the type of synoptic system affecting a region. Poor visibility condition has an ill effect on the air traffic and transportation system Fabbian et al. (2007), which can badly impact the economy Holtz & Wachs (2011). Moreover, poor visibility due to hazy weather decreases road safety and increases the risk of traffic congestion Gao et al. (2020). There are a lot of research works regarding the estimation of visibility applying machine learning algorithms (mostly regression techniques) using the meteorological data of various regions Gultepe et al. (2006); Li et al. (2017). Ortega et al. (2019) compared five different machine learning classifiers to determine the visibility in the region of Florida in USA. Castillo. B et al. (2022) explored the performance of different classification and regression techniques to predict low visibility events over a data from Mondonedo weather station at Galicia in Spain. There are many other research works for visibility prediction using machine learning and artificial neural network techniques Bari & Ouagabi (2020); Cho & Palvanov (2019). However, to our knowledge, none of these works estimate the visibility of the Indo-Gangetic plane using machine learning, which witnesses widespread fog (a term used for conditions when visibility is less than 1 km), primarily during winter. Despite the large spatial extents, the localised physical nature and spatio-temporal variations of visibility at various scales pose huge challenges in its accurate estimation. Therefore, it is crucial to analyze the factors and their impact on visibility using better data-driven approaches.

## 2 PROPOSED FRAMEWORK

A regression framework has been developed here to estimate the visibility in the Indo-Gangetic plane in two stages. Note that the meteorological data generally contain mixture of numerical and categorical features and for many features certain values may not be recorded due to system failure. In the first stage, some standard feature engineering schemes has been used to identify potential spatio-temporal features of the meteorological data. The surface meteorological data that is used in this work contain the data from different weather stations in the Indo-Gangetic region, collected from hourly observations contained in US National Oceanic and Atmospheric Administration (NOAA) surface data repository Rutledge et al. (2006). We have grouped the data of each weather station with an unique id to combine them to diminish the effect of the missing values as individual features

---

*Corresponding author

Table 1: Experimental Results of Different Methods for Visibility Estimation

| Evaluation Measure | AB | DT | ENet | LGBM | LR | LO | RF | RG | RNN | Proposed |
|---|---|---|---|---|---|---|---|---|---|---|
| MAE | 2.35 | 1.37 | 3.10 | 0.30 | 3.19 | 3.12 | 1.03 | 2.7 | 2.21 | **0.03** |
| RMSE | 7.77 | 6.07 | 8.79 | 0.45 | 8.80 | 8.80 | 4.99 | 8.10 | 1.91 | **0.06** |

of a particular weather station have less variations than the other stations. Note that the data set has lot of missing values of individual features and hence different standard schemes have been used here to deal with these missing values. The median of all the values of a particular feature is used here to replace the missing values for that feature as the median is a representative of the given feature Lin & Tsai (2020). Similarly for categorical features, mode of a feature instead of median is used to replace the missing values for that feature. Subsequently, to apply the regression techniques on this data set, categorical features are transformed to numerical values following one hot encoding scheme Seger (2018), which is an widely used scheme in this regard. XGboost method Luckner et al. (2017), a state of the art regression model is used to derive relation between different features to estimate the target variable, i.e., visibility. XGboost is an efficient and scalable implementation of gradient boosting technique Chen & Guestrin (2016), which is used here to estimate the visibility from the given data as it performed very well in similar applications Pan (2018); Li et al. (2019).

## 3 EXPERIMENTAL ANALYSIS

The dataset contains spatiotemporal data of 31 different cities in the Indo-Gangetic plane and it is collected from hourly observations contained in NOAA surface data repository Rutledge et al. (2006). The original dataset has 934807 instances from different weather stations and 121 features for each instance, out of which, 104 features were removed as they have more than 50% missing values. For rest of the features, the missing values were replaced by the median or mode of the other values of the features as stated in section 2. The experimental results are shown in Table 1 which are obtained using these 17 features of the test data. The performance of the proposed framework is compared with the state of the arts viz., linear regression (LR) Montgomery et al. (2021), decision tree (DT) Wantuch (2001), random forest (RF) Kim et al. (2021), lasso (LO) Castillo. B et al. (2022); James et al. (2013), ridge (RG) Uyanık et al. (2021), LightGBM (LGBM) Yu et al. (2021), adaptive boosting (AB) Jakhar et al. (2021), elastic NET (ENET) Castillo. B et al. (2022) and RNN based Jonnalagadda & Hashemi (2020) regression techniques in terms of RMSE and MAE Chai & Draxler (2014); Qi et al. (2020). The proposed method is iterated for 1000 times to avoid the effect of randomization. The data is randomly split into 90% as training data and 10% as test data. The training data is used to tune the parameters of individual regression techniques following 10-fold cross validation technique. Subsequently, the best set of parameters is implemented on the test data for each method. Note that a low score of MAE and RMSE indicate better performance of a regression technique than another method. Table 1 shows that the performance of the proposed method is better than the other techniques in terms of MAE and RMSE. It may be noted that both MAE and RMSE of the proposed one is almost 0, which indicate the method is working reasonably well. These results clearly show the effectiveness of the proposed framework to estimate the visibility of an area. Paired t-test Ruxton (2006) is performed on the scores of proposed framework and other methods in Table 1 to test the statistical significance. The p-value threshold is set to 0.005 here i.e., anything below 0.005 rejects the null hypothesis. It has been found that 16 out of 18 cases are statistically significant when the proposed method beats other techniques in Table 1. The results are inconclusive for the rest two cases in Table 1. Thus it can be concluded that the proposed one performs significantly better than other techniques in 88.88% (16/18) cases in Table 1.

## 4 CONCLUSIONS

The error rate of the proposed method is very low, but still there are scopes to improve the performance further by using the meteorological data of various other regions across globe. Note that the framework is evaluated based on the ground truths of the meteorological data of Indo-Gangetic region. Furthermore, no domain expert is involved in this work and no specific domain knowledge is used to build the framework. We believe that the method has the potential to be used for visibility estimation of any area across globe using the meteorological data. In future, the performance of this framework can be explored on the meteorological data of other regions for visibility estimation.

URM STATEMENT

The authors acknowledge that at least one key author of this work meets the URM criteria of ICLR 2023 Tiny Papers Track.

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
