# OpenReview forum: "AN ENSEMBLE LEARNING FRAMEWORK FOR VISIBILITY PREDICTION IN INDO-GANGETIC REGION"
_ICLR.cc/2023/TinyPapers — Submitted to Tiny Papers @ ICLR 2023_

### Official Review · Reviewer_BTMV · 2023-03-30

**Confidence:** 4

**Summary Of Contributions:**

The paper presents a regression framework for estimating visibility in the Indo-Gangetic plane using meteorological data. The authors analyze the factors and their impact on visibility using data-driven approaches and propose a two-stage process for feature engineering and regression. The paper aims to fill a gap in the literature by focusing on the performance of machine learning techniques for visibility estimation in this region.|

**Rating:**

Great Start (GS): a submission which meets some of the reviewing criteria but has room for improvement

**Strengths And Weaknesses:**

## Strengths

- The introduction provides a good overview of the importance of visibility estimation, its impact on air traffic and transportation, and the previous research works on this topic.
- The proposed framework is well-structured, including the feature engineering schemes used to deal with missing values in the meteorological data.
- The use of XGboost method for regression is appropriate as it has been shown to perform well in previous applications.
- The experimental analysis section presents the dataset used in the study, the evaluation measures used, and the results of different regression methods for visibility estimation.

## Weaknesses

- The introduction could be improved by providing more specific information on the challenges of visibility estimation in the Indo-Gangetic plane and how the proposed framework addresses them.
- The experimental analysis section could be expanded to include a more detailed analysis of the results, such as statistical significance tests and visualization of the predicted vs actual visibility values.
- The limitations and potential future directions of the proposed framework are not discussed.
- The author briefly mentions using "standard feature engineering schemes" to identify potential spatiotemporal features, but does not provide any detail on what these schemes are or how they were implemented. Similarly, the author mentions using "different standard schemes" to deal with missing values, but does not specify what these schemes are or provide any justification for their use. Providing more detail and explanation of these methods would improve the clarity and reproducibility of the work.
- Although the research question is important and the methodology is sound, the findings do not present any new or groundbreaking insights into the topic. The paper largely relies on existing literature and does not contribute any new ideas or perspectives. For example, in the introduction section, the author briefly mentions previous studies on the topic and outlines their findings. However, the author does not indicate how their study differs from these previous works or how it adds to the current body of knowledge. This lack of novelty in the paper could limit its potential impact and relevance in the field.
- This paper also does not compare to any latest tabular deep learning methods.

**Suggested Changes:**

- This paper does not contain a URM statement (also a note to the program commitee)
- I would most certainly suggest having an abstract, I understand that the limit is merely two pages however, I would suggest still finding some way to put in a short abstract maybe by moving some content to an Appendix
- Multiple incorrect citations throughout the text mainly due to:
  - using in-text citations when parenthetical ones were expected
  - duplicating citations one after the other

Here are a few examples from the text:

> Poor visibility condition has an ill effect on the air traffic and transportation system Fabbian et al. (2007), which can badly impact the economy Holtz-Eakin & Wachs (2011). Moreover,
poor visibility due to hazy weather decreases road safety and increases the risk of traffic congestion
Gao et al. (2020).

>  using the meteorological data of various regions Gultepe et al. (2006); Li et al. (2017). Li et al. Li et al. (2017) developed an advanced
numerical prediction model for detecting the factory emission which were further used to build a visibility prediction system. Ortega et al. Ortega et al. (2019) compared five different machine learning classifiers to determine the visibility in the region of Florida in USA.

- The "Introduction" section in the text combines "Related Works", I would suggest having separate sections for these.
- Some sentences in the paper lack clarity:
> The increase in fog and poor visibility has become prominent over the last 50 years in this
region.

Do you show this in your paper, if so make that abundantly clear and if not include a citation

- I would suggest removing any repetitions considering the short page limit

> The dataset is collected from hourly observations contained
in NOAA surface data repository Rutledge et al. (2006).

- The methods employed should be made clearer, what features were removed, what features are now left, how do you identify that features constitute enough impact on the target.

> Out of these 121 features, 104
features have been removed as they have more than 50% missing values, but for rest of the features,
the missing values were replaced by the median or mode of the other values of the features as stated
in section 2.

- I would suggest comparing your methods to other state-of-the-art tabular data methods instead of the naive machine learning methods compared with, in this paper.

---

### Official Review · Reviewer_nTju · 2023-04-01

**Confidence:** 3

**Summary Of Contributions:**

The paper compares the performance of several regression methods on the task of prediction of the visibility from weather data.

**Rating:**

Needs Clarification (NC): a submission which does not meet the reviewing criteria and needs clarification for its described problem or solution

**Strengths And Weaknesses:**

Strength:
- Several regression methods are used in the experiment

Weaknesses:
- The paper does not propose any novelties. It is not clear what the contribution of the proposed framework is, is it just using XGBoost for the task?
- The performance gap between XGBoost and other methods is large. However, it is not clear what is causing the huge gap.
- It would be good if other DL-based prediction methods were used for the comparison. What would be the results with a simple MLP?
- A detailed description of the dataset features is missing. What are the 17 features and their value ranges? How did you handle time features? Did you use normalization?


**Suggested Changes:**

I would suggest the authors add a detailed description of their dataset and its features. I encourage using other prediction methods as well as the regression methods used. Finally, a discussion on reasons for the good performance of XGBoost is missing.

---

### Meta-Review · Area_Chair_P38f · 2023-04-07

**Recommendation:** Invite to archive
**Confidence:** 4

**Metareview:**

This paper does not contain a URM statement.

The paper noted the gap in the existing literature on estimating visibility in a specific region and attempt to fill this gap by applying XGboost classifier to existing meteorological data. Though the methods seem interesting and fill the gap in existing literature, reviewers agree that the paper has limited novelty and lacks a discussion on the description of their dataset, its features selection, and comparison with existing tabular deep learning methods.

**Summary:**

The main contribution of the paper is to show the use of XGboost techniques to estimate the visibility of the area using meteorological data in the presence of occlusions, such as Fog. The authors applied this technique to estimate the visibility of the Indo-Gangetic plain, which the existing model lacks. However, the lack of details on datasets and feature selection and the selection of XGboost makes is hard to judge the novelty of this work.

**Reason For Not Giving A Higher Recommendation:**

The paper lacks on describing the proposed technique in detail and miss important comparison against high-performing deep network.

**Reason For Not Giving A Lower Recommendation:**

NA

---

### Decision · Program_Chairs · 2023-04-10

**Decision:**

Invite to archive

**Comment:**

Please be sure to include an URM statement in the revised archived version. Thank you.